# Ideas and perspectives: Alleviation of functional limitations by soil organisms is key to climate feedbacks from arctic soils

Gesche Blume-Werry[1], Jonatan Klaminder[1], Eveline J. Krab[1,2], Sylvain Monteux[3,4]

[1]Climate Impacts Research Centre, Department of Ecology and Environmental Science, Umeå University, Sweden
[2]Swedish University of Agricultural Sciences, Department of Soil and Environment, Sweden
[3]Department of Environmental Science, Stockholm University, Sweden
[4]Bolin Center for Climate Research, Stockholm University, Sweden

*Correspondence to*: Sylvain Monteux (sylvain.monteux@aces.su.se)

**Keywords**: arctic, decomposition, detritivores, soil fauna, tundra

**Abstract.** Arctic soils play an important role in Earth's climate system as they store large amounts of carbon that, if released, could strongly increase greenhouse gas levels in our atmosphere. Most research to date has focused on how the turnover of organic matter in these soils is regulated by abiotic factors and few studies have considered the potential role of biotic regulation. However, arctic soils are currently missing important groups of soil organisms and here, we highlight recent empirical evidence that soil organisms' presence or absence is key to understanding and predicting future climate feedbacks from arctic soils. We propose that the arrival of soil organisms into arctic soils may introduce 'novel functions', resulting in increased rates of *e.g*., nitrification, methanogenesis, litter fragmentation, or bioturbation, and thereby alleviate functional limitations of the current community. This alleviation can greatly enhance decomposition rates, in parity with effects predicted due to increasing temperatures. We base this argument on a series of emerging experimental evidence suggesting that the dispersal of until-then absent micro-, meso- and macro-organisms (i.e., from bacteria to earthworms) into new regions and newly-thawed soil layers can drastically affect soil functioning. These new observations make us question the current view that neglects organism driven 'alleviation effects' when predicting future feedbacks between arctic ecosystems and our planets' climate. We therefore advocate for an updated framework in which soil biota and the functions by which they influence ecosystem processes, become essential when predicting the fate of soil functions in warming arctic ecosystems.

## 1 Introduction

Arctic soils store close to half of worldwide soil carbon (Hugelius et al., 2014; Strauss et al., 2017) and the potential feedbacks between the about 1300 Pg-C stored in arctic soils and our planets climate system are causing concern (IPCC, 2021). To date, the prevailing view is that low temperatures are a primary control of this carbon store, especially of the 822 Pg-C stored frozen in permafrost. This view is well-supported by studies highlighting the top-down control of temperature, next to substrate

quality and oxygen availability, on microbial processes (Conant et al., 2008; Razavi et al., 2017). It is also well-established that the widespread presence of permafrost, a soil feature closely linked to temperature, is currently constraining decomposition in arctic soils (Goulden et al., 1998). However, arctic soils currently lack many species of soil organisms that are key drivers

of decomposition at lower latitudes (Hodkinson and Wookey, 1999; Golovatch and Kime, 2009; Sfenthourakis and Hornung, 2018; Briones, 2014; Aerts, 2006), potentially leaving open niches. As decomposition processes can be driven by the functional dissimilarity among the decomposers present (Heemsbergen et al., 2004) and the matching of traits between decomposers and available resources (Lustenhouwer et al., 2020), these open niches imply that the absence of certain decomposer soil fauna (such as woodlice, millipedes or geoengineering earthworm species) but also microbial decomposers may hamper

decomposition rates of soil organic matter in the Arctic. We here refer to this idea as 'functional limitation' and use 'soil organism functions' as the direct effect of an organisms' activity (via its combined functional or 'effect' traits) on soils, litter, or substrate (e.g. the ability or extent to which they perform nitrification, methanogenesis, litter fragmentation, microbivory, or bioturbation). As a consequence of these open niches and functional limitations, the arrival of these organisms with additional functions, for example those that stimulate bioturbation of deeper soil layers or induce high litter fragmentation

rates, would increase the functional diversity and could greatly stimulate decomposition.

To what extent such functional limitations in arctic soils in the past contributed to the build-up and persistence of large carbon pools it not well known. In fact, there is no general consensus if the functionally-limited soil communities of arctic soils are a result of the harsh climate, or simply due to slow northward dispersal rates of certain soil organisms after the last glaciation. Based on the current distribution of soil fauna in the northern hemisphere, climatic conditions do indeed seem to act as a prime

regulator of soil organisms and their functions(Golovatch and Kime, 2009; Kuznetsova and Gongalsky, 2012; Maynard et al., 2019; Sfenthourakis and Hornung, 2018). Nevertheless, the presence of species' functions can also be shaped by glacial history (Mathieu and Davies, 2014), and studies suggest that the absence of some soil fauna species in the Arctic is rather due to limited natural dispersal vectors than present day environmental constrains and that large areas of the Arctic might be suitable for establishment of certain decomposing soil organisms already now or in the near future (Blume-Werry et al., 2020; Coulson,

2015; Wackett et al., 2018).

In this opinion piece, we propose that increasing temperatures are opening up new niches for soil organisms in arctic soils, both laterally and vertically, and that the effect which newly arriving organisms may have on decomposition processes could be substantial. This is based on the knowledge that soil organisms are important components of the decomposition process everywhere (Lavelle, 1997; García-Palacios et al., 2013; Griffiths et al., 2021) and for example earthworms, millipedes,

isopods, and collembola all can substantially increase mass loss or $CO_2$ emissions (e.g., (Addison and Parkinson, 1978; Cárcamo et al., 2000; Des Marteaux et al., 2020), especially if functionally diverse species combinations are present (Heemsbergen et al., 2004; Delgado-Baquerizo et al., 2020). Moreover, we highlight that slow historic dispersal of soil fauna can, at least partly, explain their current absences in arctic soils. Soil macro-fauna in particular disperses at slow rates, thus that there is a time-lag, so called 'invasion debt' (Rouget et al., 2016), before soils that were previously constrained by

glaciation or frozen soils develop food-webs that contain all major functions. Our perspective, illustrated in Fig. 1, introduces

a framework stating that arctic soils are currently in a functionally-limited decomposition stage that could be alleviated by lateral, northward dispersal into currently unoccupied areas as well as by vertical (downward) dispersal of novel soil organisms into newly-thawed soil layers. This implies that once soil organisms with currently missing functions arrive and more complex food webs develop, decomposition rates may be much higher than suggested from warming of contemporary tundra soils alone (Aerts, 2006; van Geffen et al., 2011; Heemsbergen et al., 2004; Frouz, 2018). In that case, models based on assumptions of how contemporary arctic soils respond to climatic variables may fail to foresee important future shifts in tundra soil processes that would arise when soil organisms with functions central for decomposition processes settle. In this perspective paper we highlight data showing that some soil organism driven functions are absent from arctic soils and consequences of introduction of species with these missing functions. Thereto, we provide examples of experiments with additions of soil organisms to estimate the impacts of novel soil organisms arriving in arctic soils on the current, 'functionally-limited' decomposition rates.

## 2 A dispersal constrained community of soil organisms in arctic soils

Soils are often considered to harbor most functions, due to the omnipresence and large diversity of soil organisms and the generally large functional redundancy assumed amongst them (Nannipieri et al., 2003). While the assumption of functional redundancy in soils is now questioned by soil ecologists, many scientists still generally assume that soil functioning is primarily determined by its physical and chemical composition and thus, that organisms are simply just there if right physiochemical conditions are met. For example, estimates of the climate feedback from arctic soils (e.g., Koven et al., 2015; Schuur et al., 2015), rely strongly on incubation studies. Implicitly, this assumes that the incubated microbial and faunal communities carrying out decomposition processes are functionally representative of the communities present in the field after thawing and in warming soils. However, arctic soils and particularly permafrost soils are likely to deviate from this assumption. Permafrost soils are indeed not only deprived of most viable fauna – although on rare occasions it has been possible to isolate viable animals such as nematodes (Shatilovich et al., 2018) or rotifers (Shmakova et al., 2021) as well as plants (Yashina et al., 2012) – but also of numerous microbial taxa, resulting in distinct microbial communities (e.g., Johnston et al., 2019; Monteux et al., 2018). Although their topsoil counterparts may have varying levels of diversity (e.g., Fierer et al., 2012), permafrost microbial communities typically exhibit low diversity, as they are shaped by strong environmental constraints over long time-scales and extreme dispersal limitation due to their frozen environment (Bottos et al., 2018; Ernakovich et al., 2022). Similarly, the biogeographical history of the Arctic, including glaciations, effectively eradicated certain groups of soil organisms from the non-frozen topsoil as well (Briones, 2014). In other words, due to the past and current environmental filtering of inland ice-sheets and frozen soils, few would argue against the view that arctic soils are unique in their current lack of micro-, meso- and macro-organisms that are present in most other soils. It is also likely that warming soils, including thawing permafrost, will open numerous new niches for such soil organisms to establish as the physical barrier of frozen soils is removed and thus far fauna-free soils can be colonized. Extrapolating ecological theory outlined for temperate or boreal ecosystems is thus not straightforward, due to the absence of entire clades or even kingdoms in some arctic soil environments (Briones, 2014).

Palaeoecological reconstructions have shown that plants have a remarkable capacity to rapidly, *i.e.,* on a decadal time-scale, colonize formerly glaciated areas (Nota et al., 2022), but less is known about the colonization rate of soil organisms after deglaciation or permafrost thaw. From studies of glacier forelands, where soil organisms can establish in open niches via short-range dispersal, we know that mature soil fauna communities can establish within a century (Kaufmann et al., 2002). However, rates of long-range dispersal across hundreds of kilometers into arctic soils are unknown and likely much lower. It has been suggested that earthworms disperse naturally with a rate of 5 to 20 m yr$^{-1}$ (Chkrebtii et al., 2015; Wackett et al., 2018; Cameron et al., 2008; Cameron and Bayne, 2015) and that this slow dispersal from glacial refugia can explain their absence in previously glaciated American forest and the Arctic. These slow dispersal rates of earthworms are likely an important factor constraining their presence, considering that several species can survive and establish in arctic soils once introduced by humans (Blume-Werry et al., 2020; Wackett et al., 2018). Similarly, several introduced species of Collembola (Coulson, 2015; Enríquez et al., 2019), tapeworms and mites (Coulson, 2015) have been shown to thrive under arctic conditions, further indicating that these species were not constrained by the arctic climate *per se*, but rather by their ability to access tundra soil by their own means. In contrast, large surface-dwelling animals, such as millipedes (Golovatch and Kime, 2009) and isopods (Sfenthourakis and Hornung, 2018) follow distribution patterns in the Arctic that suggest temperature itself limits their range, rather than the glaciation history. However, this distribution only suggest that their dispersal is fast enough to colonize niches in the Arctic over Holocene time-scale and thus, it is not self-evident that they can respond at time-scales of relevance for the ongoing climate change, *i.e.* centennial time-scale, unless introduced by humans.

Microbial dispersion into Arctic soils functions in a different way than faunal dispersal. Here, lateral, northward dispersal is likely less limiting than for soil fauna because airborne dispersal is widespread in many bacteria and fungi (Harding et al., 2011; Thompson et al., 2017), but the vertical dispersal of micro-organisms into newly-thawed layers is the subject of ongoing investigations. While microbial communities in newly-thawed permafrost can converge with those observed in the active layer (e.g. Monteux et al., 2018; Doherty et al., 2020), it remains unclear whether this stems from microbial migration downward or from modifications of the existing permafrost microbial community. What further complicates predictions of future microbial communities in the Arctic is the existence of ancient bacteria and viruses in permafrost which, after being dormant for millennia in frozen soil layers, can become active again upon thaw (Miner et al., 2021). How the active and dormant microorganisms currently present in permafrost will interact with newly-arriving microorganisms to determine the assembly of post-thaw permafrost microbial communities is still unclear (Ernakovich et al., 2022).

**3 Evidence for alleviation of functional limitation with novel soil organisms in the Arctic**

If contemporary, or near-future, climatic conditions in arctic soils do allow novel soil organisms to establish in previously 'functionally-limited soils', it is highly relevant to assess how soil organic matter turnover may change when soil organisms with missing functions arrive and more niches are filled. It has been shown several times (e.g., Wall et al., 2008; García-Palacios et al., 2013) that larger soil organisms had limited influence on decomposition processes in arctic soils compared to

other areas. Yet, such studies are inherently limited by the simplified food web present in arctic soils right now and cannot account for the potential contribution to the decomposition process of soil fauna species that are currently absent (Frouz, 2018). For example, when litterbags with subarctic leaf litter were incubated in other ecosystems, access of larger soil fauna did increase mass loss (Makkonen et al., 2012). That is, the same mesh size does not exclude or include the same soil macrofauna in northern sites as elsewhere. Estimating the effect of an alleviation of functional limitation through filling presently empty

niches, therefore, requires the experimental addition of soil organisms to achieve a complex soil food web without missing functions and empty niches. Indeed, several recent studies suggest that the additions of soil organisms with novel functions can have a substantial impact on arctic soil organic matter cycling (Blume-Werry et al., 2020; Monteux et al., 2022, 2020; Marushchak et al., 2021). Below, we exemplify that soil organisms on the micro-, meso-, and macro-scale can alleviate functional limitations and by doing so have profound consequences on arctic plant communities and biogeochemical cycling.

Earthworms are probably the best-known example of invasive soil macrofauna. They are incredibly powerful ecosystem engineers that alter the physical and biogeochemical properties of the soil through increased litter decomposition and soil mixing (Fahey et al., 2013) and change soil microbial and faunal communities (Ferlian et al., 2018), thereby affecting ecosystem functioning and ultimately plant communities (Craven et al., 2017; Mathieu et al., 2022). Though most focus of earthworm research has been on invasions in North American temperate and boreal forests, these processes are likely very

relevant in arctic soils as shown in pioneer surveys (Wackett et al., 2018). Geoengineering, *i.e.* endogeic and anecic, earthworms are generally absent from arctic soils, but have been found in isolated patches across the Arctic where they can not only survive but spread out after human introduction (*e.g.,* Blume-Werry et al., 2020; Tiunov et al., 2006; Wackett et al., 2018). Geoengineering earthworms are known to rapidly deplete thick organic layers in boreal forests through increased decomposition and mixing (Lejoly et al., 2021), likely resulting in carbon release to the atmosphere (Fahey et al., 2013),

making future earthworm-induced greenhouse gas emissions a concern for arctic soils. Indeed, litter decomposition and organic matter turnover seem to be stimulated immensely when earthworms arrive in tundra soils. Blume-Werry et al. (2020) showed in an earthworm addition experiment into tundra mesocosms that geoengineering earthworms rapidly and substantially increased plant nitrogen content and plant growth above- and belowground in different tundra plant communities. Late season root growth in the first year of the experiment, for example, was almost twice as high when earthworms were present. Changes

in vegetation greenness and nitrogen concentration were even of a similar magnitude or larger, respectively, than 3 °C of warming (Blume-Werry et al., 2020). In subarctic microcosms, earthworm addition increased both litter mass loss and $CO_2$ fluxes, as did the addition of other macrofauna generally absent from subarctic ecosystems, i.e. isopods and millipedes (van Geffen et al., 2011). Thus, upon the arrival of novel soil macrofauna, such as but not limited to earthworms, new functions seem to be introduced in the soil food web that remove current bottlenecks in organic matter turnover, with thus far

unquantified consequences for carbon and nutrient cycling.

    Soil mesofauna, such as collembola, nematodes, rotifers, and tardigrades, affect litter decomposition rates through their feeding activities by inoculating litter with microbes, increasing the surface area of litter substrates, and increasing overall microbial activity through selective grazing (Beare et al., 1992; Lussenhop, 1992). They thus also represent organisms potent enough to

cause substantial impacts on soil processes when they colonize new areas or soils. In a mesocosm study mimicking a drained thermokarst or thaw slump scenario, Väisänen et al. (2020) found that within a year microarthropods –but not enchytraeids– were able to settle into newly-thawed permafrost at densities one order of magnitude below those found in the surrounding active layer. Monteux et al. (2022) assessed how mesofauna, collembola, affect carbon dioxide emissions from newly-thawed permafrost soils. Collembola are ubiquitous throughout the Arctic and can be found at very high densities (e.g., 130 000 individuals per square meter in high arctic Greenland, Sørensen et al., 2006), and are therefore likely to colonize newly-thawed permafrost where it is not water-saturated. In the study by Monteux et al. (2022), carbon dioxide production from permafrost soils increased by 26% when collembola were present. While about half of this effect could be attributed to collembola respiration itself, the remaining 13% directly resulted from increased soil organic matter decomposition. Presence of collembola also increased $CO_2$ emissions from topsoils by up to 400% in a high arctic site (Addison and Parkinson, 1978). These findings imply that standard incubation studies of permafrost or active layer soil without additional soil fauna, could be strongly underestimating the potential carbon emissions of these soils upon thawing.

Not only macro- and mesofauna are absent in permafrost soils, also their microbial communities differ from those found in the overlying active layer (Doherty et al., 2020; Johnston et al., 2019; Monteux et al., 2018) and the functional potential of these specialized microbial communities for decomposition processes can also be drastically smaller than that of active layer communities. Consequently, if new microorganisms are added to thawed permafrost soils, they can alleviate functional limitations and strongly increase carbon dioxide production (+38%, Monteux et al., 2020), but also initiate methanogenesis (Knoblauch et al., 2018) or nitrification (Monteux et al., 2020). In other words, such ecosystem processes seem not limited by the lack of adequate substrates, but rather by the absence of microorganisms harboring the specific genes needed to carry out these biochemical transformation processes. These findings are not constrained to laboratory incubations but can also be observed in more realistic field settings. For instance, nitrogen cycling gene abundances and process rates are very low immediately following permafrost thaw in Yedoma exposures, but substantially increase with ecosystem complexity as new functions are introduced by newly-arriving organisms (Marushchak et al., 2021). These new functions increased $N_2O$ production by 1 to 2 orders of magnitude, an effect which would be omitted by incubation studies focusing solely on the functionally limited microbial communities present in permafrost before thaw. Taken together with similar findings on methane and carbon dioxide production (Knoblauch et al., 2018; Monteux et al., 2020), it seems evident that introduction of microbes with novel functions or increased efficiency can boost the emission of several greenhouse gases from thawing permafrost. The dynamics of microbial community assembly upon permafrost thaw are a growing field of research (see Ernakovich et al., 2022), and we advocate for further effort in exploring how assembly dynamics affect functional limitations, their possible alleviation, and interactions between microbial and faunal communities. Mechanistic studies could explore which functions are missing from permafrost microbial communities, such as proxies of decomposition, nutrient cycling, or production of greenhouse gases. This could be achieved in incubations by manipulating permafrost microbial communities using 'positive controls' to explore if there are functional limitations present and apply active layer microorganisms to test if they are able to alleviate these functional limitations (similar to Monteux et al., 2020).

## 4 The contemporary mismatch between climate, plants, and soil organisms

Arctic regions are unique in several ways. They are shaped by strong environmental filters, both in the past and present, which resulted in unmatched high allocation of plant biomass belowground relative to aboveground (Fig. 2a) and soil organisms adapted for survival rather than high functional performance (Crowther et al., 2019; Nielsen and Wall, 2013). Low functional performance and the resulting limited decomposition rates have led to a large build-up of soil organic matter in arctic and boreal soils, as illustrated in Fig. 2a. This figure was created from a multitude of data sources: above- and belowground plant carbon stocks for 2010 are from Spawn et al. (2020), total soil organic carbon stocks are the sums over 0-2 m depth from SoilGrids250m 2.0 (Poggio et al., 2021); current and future permafrost stocks by Keuper et al., (2020) are obtained from applying CLM4.5 simulations (Koven et al., 2015) to SOC stocks from NCSCDv2 (Hugelius et al., 2014). Permafrost data are for deposits between 0-3 m and thus exclude about half of permafrost SOC contained in deeper deposits (Strauss et al., 2017; Hugelius et al., 2014). However, as increasing temperature and changes in snowfall patterns are rapidly changing the arctic environment above- and belowground, new niches and opportunities are opening up for soil organisms to utilize the large energy sources stored at depth in arctic soils. New niches can arise both through direct climate changes, or indirectly through vegetation changes (Kaufmann et al., 2002; Krab et al., 2019) which are widespread throughout the Arctic (Elmendorf et al., 2012; Myers-Smith et al., 2019). Sound projections about the future fate of soil carbon in arctic soils thus depend on correct understanding of processes controlling decomposition in the near future. We identified two different, simplified scenarios based on existing knowledge. The **first scenario**, which seems to be the theory most studies apply, depicts the 'state-of-the-art' (Scenario 1, Fig. 2b). Here, large-scale and dramatic changes in the belowground environment do not lead to a change in the presence or depth-distribution of soil organisms. This conceptual view assumes that the fate of arctic soils C can simply be predicted by combining *in vitro* incubation studies, field observations, and modelling of soil and plant responses. Such simplified assumptions have been paramount in providing estimates of the permafrost carbon feedback (e.g., Koven et al., 2015) and some plant-soil interactions (Keuper et al., 2020). However, while warmer soils alone can increase activity and turnover rates of soil organisms, for example through an increase in density (Dollery et al., 2006), we assume that significant changes in the functional potential of the soil organisms only occur with changes in community composition (Crowther et al., 2019). Thus, our **second scenario** (Scenario 2, Fig 2b) highlights that the current soil food-webs might not be representative for the future and that the impacts of new functions, currently absent in the contemporary environment, need to be accounted for. Future arctic soils may have a more functionally diverse soil community (Scenario 2, Fig. 2b) in which new soil organism functions arrive, both in topsoils and in deeper soil layers, and thus increase the rates of decomposition processes (Heemsbergen et al., 2004). As outlined in this perspective piece, there are numerous studies in support of his second scenario making it highly relevant to account for northward dispersion of soil fauna in future models. For example, woodlice distribution seems to be restricted south of the limit of 120 days per year with a temperature above 10°C (Sfenthourakis and Hornung, 2018), and might thus progress northwards along with that limit. Likewise, millipedes appear absent from most regions affected by permafrost (Golovatch and Kime, 2009) and therefore might be able to disperse northwards once permafrost recedes. As

macro-decomposers breaking down large litter elements into smaller pieces, woodlice and millipedes provide important ecosystem functions to soil and can significantly speed up decomposition (Joly et al., 2018; Lavelle, 1997), and their dispersal into soils where they are absent could therefore affect their biogeochemical cycling. Interestingly, Golovatch & Kime (2009) also show isolated millipede occurrence outside of their regular distribution range, illustrating that they can indeed survive beyond their current distribution range already now. This suggests that their current distribution does not reflect a simple thermal niche as the current climate does not fully explain the absence of millipedes and dispersal limitation might be more important. Similarly, geoengineering earthworms are also mostly absent from previously glaciated areas but are successfully settling at and dispersing from points of anthropogenic introduction. Again, this suggests that the biogeographical history will play a smaller role in the future, as they will colonize more and more of these systems where they were until now absent (Blume-Werry et al., 2020; Wackett et al., 2018). If and how fast which groups will reach arctic soils, whether on its own of via anthropogenic dispersal, is difficult to assess as there is an overall lack of data on dispersal abilities (Aerts, 2006; Hickling et al., 2006; David and Handa, 2010). Even the limits of spatial distribution are poorly defined for several important groups of soil fauna groups, as the northernmost range of their apparent distribution coincides with areas where samplings are scarce, and it is often not clear whether a given study did not find such or such group or did not look for it (Bastida et al., 2020; Lavelle et al., 2022).

Of the belowground changes in the Arctic, the widespread thawing of permafrost (Smith et al., 2022) is probably the most striking as it removes an obvious barrier for soil fauna dispersal and opens up new habitats. This means that there might not only be more species with new functions in the topsoil but also an increase in functionality deeper down in the newly-thawed soils (Fig. 2b). The thawing of permafrost happens overall as a thickening of the seasonally thawed layer above the permafrost, upon which the new soil volume is explored by roots of certain plant species (Blume-Werry et al., 2019; Finger et al., 2016). Microbial communities in the newly-thawed permafrost become similar to the active layer communities (Doherty et al., 2020; Monteux et al., 2018), although it is unclear to what extent this stems from downwards dispersal, influence of plant roots, or endogenic changes from the permafrost communities. While soil meso- and macrofauna are less likely to substantially colonize these deep, often water-logged thawing layers, permafrost does not solely thaw as a thickening of the active layer. Various thaw features can be observed, such as drained thermokarst, retrogressive thaw slumps or active layer detachments (Inglese et al., 2017; Olefeldt et al., 2016). In these circumstances, former permafrost becomes thawed and exposed to surface conditions, thus providing suitable new habitats for soil organisms. At least micro- and mesofauna appear able to establish in this newly-thawed permafrost (Väisänen et al., 2020), with hitherto unclear consequences, although their impact on bacterial community composition seems rather limited (Monteux et al., 2022, this special issue). While our proposed scenarios are simplified, and may be further complicated by e.g., rewiring of interactions (Woodward et al., 2010), non-linear responses (Fox et al., 2006), or novel species interactions (Gilman et al., 2010), we believe that establishing studies that include and manipulate soil fauna are a necessary step to better predict feedbacks from arctic soils to the climate.

## 5 Conclusions

Here, we postulate a contemporary mismatch between climate changes, plant responses, and colonization by soil organisms across the Arctic, leading to a currently functionally-limited decomposition. If the complexity and function of the food web are not explicitly manipulated, a potential future functional alleviation is not captured by warming experiments thus inadvertently missing out on essential system shifts. We thus advocate for improved and accessible data on distribution of functional groups of soil decomposers, notably macro-detritivores and geoengineering earthworms, in the circum-arctic region Ideally, this data would include whether populations out of their apparent climate range are relict or human-introduced, and experiments specifically testing the effects of alleviation of functional limitations by one or more of these functional groups such that the scientific community can better predict the true feedback potential from arctic soils to the global climate.

### Author contributions

GBW conceived the idea based on discussions with SM, JK, and EJK. GBW led the writing of the manuscript with substantial input from JK, EJK and SM. EJK was financially supported by VR (grant 2021-04458). All authors contributed critically to the drafts and gave final approval for publication.

### Conflict of interest

The authors declare that they have no conflict of interest.

### Data availability

No primary data was used in this manuscript, data shown in Fig 2a was retrieved from Keuper et al 2020 (https://git.bolin.su.se/bolin/keuper-wild-2020), Spawn et al., 2020 (https://daac.ornl.gov/cgi-bin/dsviewer.pl?ds_id=1763) and Poggio et al., 2021 (https://files.isric.org/soilgrids/latest/).

**Figures**

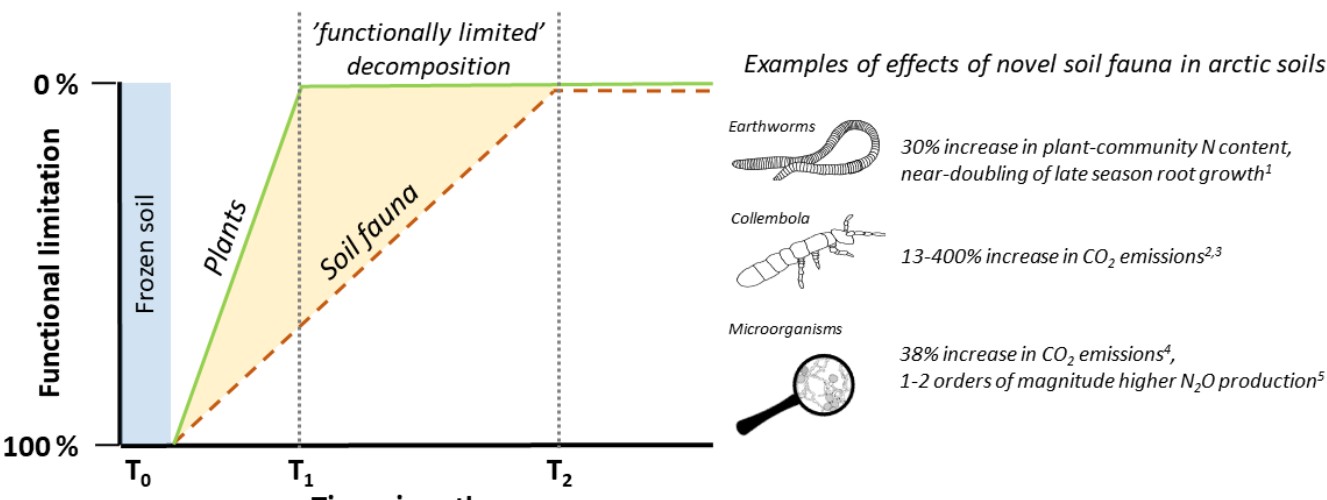

**Figure 1. Conceptual illustration of the theory underlying this perspective paper.** Following the retreat of glacier ice or thawing of permafrost, soil processes previously constrained by cryogenic processes are alleviated. At this point in time ($T_0$), novel plants and soil biota can establish in previously unoccupied areas or soil layers. As plant niches are expected to be filled at a higher rate, all possible groups of plant functional traits are represented at time-stage $T_1$ while some of soil organism functions arrive later ($T_2$). During the outlined scenario soils evolve between time interval $T_2$-$T_1$ with a 'functional limitation', *i.e.* where key functions in the food web may be missing. Here, groups of organisms with specific functions may be lacking, not necessarily because of climatic drivers but possibly due to slow dispersion vectors. We propose here that arctic soils are currently in T1-T2, detritivore-limited decomposition, implying that once they arrive and fully complex food webs develop, decomposition rates will be much higher than suggested from warming of contemporary tundra soils alone. On the right side we highlight examples of effects of novel soil fauna in arctic soils as discussed in section 3; [1]Blume-Werry et al. (2020), [2]Monteux et al. (2022), [3]Addison and Parkinson (1978), [4]Monteux et al. (2020), [5]Marushchak et al. (2021.

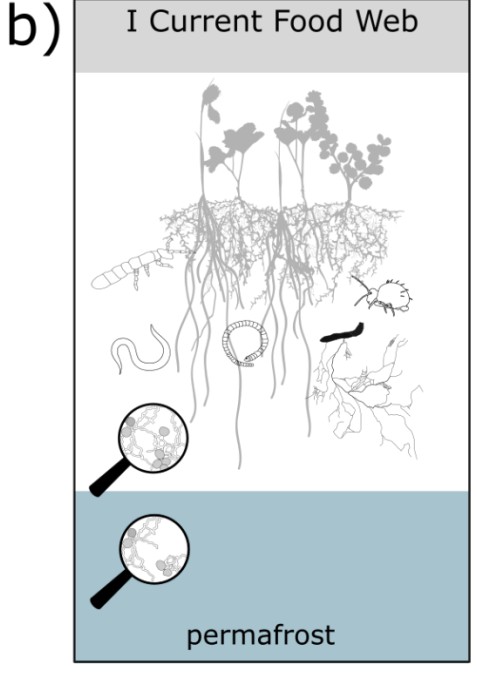

a)

b)

| I Current Food Web | II Future Scenario 1, 'state-of-the-art' | III Future Scenario 2, 'functional alleviation' |
|---|---|---|

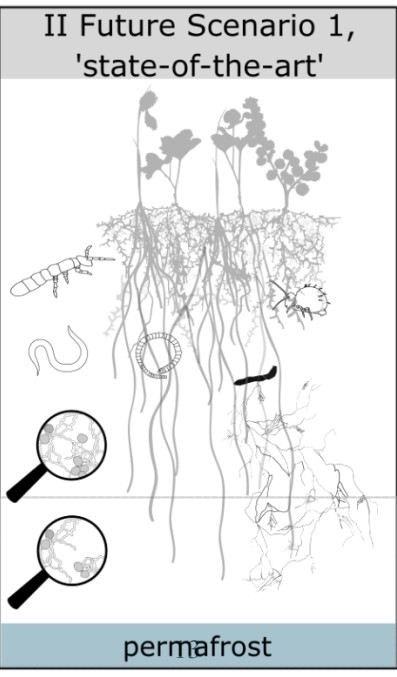

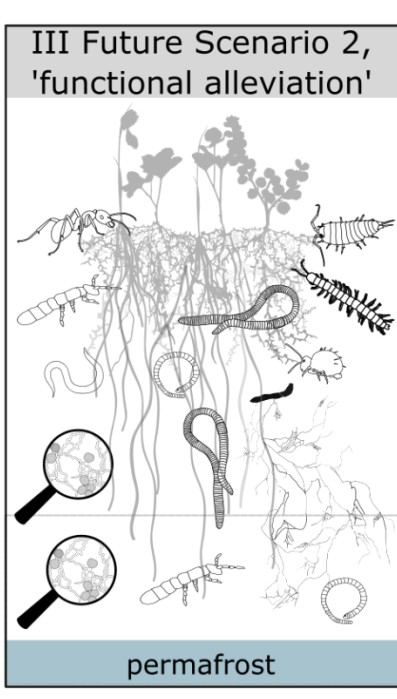

**Figure 2**.Arctic and boreal soils are characterized by disproportionately large amounts of belowground plant biomass and large stores of soil organic carbon. Climate change opens up new habitats both in latitude and depth as soils thaw and warm up, but current predictions assume no accompanying changes in the soil fauna and decomposition process. a) Latitudinal distribution of soil carbon, plant biomass above- and belowground. This figure was inspired by Fig. 2 c) in Crowther et al. 2019, but created with updated data sources and including currently and future frozen carbon pools (see main text)  b) Conceptual illustration of

the current, functionally limited, arctic soil food web and two future scenarios, 'state-of-the-art' and 'functional alleviation'. The current food web in arctic topsoils (I, Current Food Web) is characterized by dominance of micro- and mesofauna such as nematodes, enchytraeids and collembola and the soil matrix in which they live is often constrained vertically by permafrost. In II a 'state-of-the-art', the same micro-and mesofauna continue to dominate the soil matrix despite vertical expansion of the soil matrix due to permafrost thaw. In III, the 'functional alleviation' scenario, functionals are added to the foodweb in both

both top- and lower soils through the establishment of novel soil organisms .

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
