# Peer review of "Ideas and perspectives: Alleviation of functional limitations by soil organisms is key to climate feedbacks from arctic soils"

_Biogeosciences, 2022_

## Author Response (AR3)

**Response to Editor**

Public justification (visible to the public if the article is accepted and published):

Dear Sylvain Monteux and co-authors,

First of all, thanks for submitting your paper entitled "Ideas and perspectives: Alleviation of functional limitation by soil organisms is key to climate feedbacks from northern soils" to our special issue. As you might have noticed, the two reviewers made a very detailed and thorough revision, and identified a number of 'elements' that could/should be improved. So far I appreciate the efforts put in the reply letter but now, in order to make a final decision on the MS's suitability for publication, I would like to see how these clarifications and revisions are addressed and how the novelty aspects are emphasized.

RESPONSE: Thank you for giving us the opportunity to revise our manuscript. We believe we have been able to address all the criticism of the reviewers and now submit the improved, revised version. Below you find our responses to yours and the reviewers' comments.

We would also like to point out that we have further thought about Reviewer 2's comment to "*Clearly differentiate organisms' "traits" and "properties""*, and have decided to avoid confusion and misinterpretation by no longer using the word 'trait' for which some ecologist use a rather narrow definition. Instead, we now exclusively talk about functions and define our usage in the first paragraph of the introduction in ll. 40-43.

Additional comments:

- I am particularly interested in how the fact that "the manuscript only brings little novelty (R#1)" is addressed in the new version of the MS. The authors provide a list of reasons against this statement. I would highlight why the Arctic is so 'unique' and why earthworms are not 'enough', which would bring some depth to this idea-perspective piece.

RESPONSE:

Yes, we have adapted our manuscript text to make these concepts clear. In short, we have:

1) More clearly emphasized that the focus in our paper is on Arctic soils in contrast to previous works on similar themes in temperate and boreal systems, by replacing 'northern soils' in the title and elsewhere in the text with 'arctic soils' or 'tundra soils'. In retrospect, we realize that the use of 'northern soil' in the previous version of the manuscript might have been slightly misleading as readers could erroneously assume we referred to more studied temperate and boreal soils.
2) Highlighted that our study, in contrast to other published work arguing for a biological control of decomposition processes, provide empirical support to our statements. Note that these findings are recently published and have not been compiled in this context before.
3) Emphasized the uniqueness of the Arctic and that studies on other organisms than 'just' earthworms are needed in the abstract line 16 and the manuscript text (lines 92-94, 153-155 and 196-198)

4) Added a summary of the experimental evidence we discuss in Fig. 1, to highlight the massive effects of added soil biota on C- and N-cycling in Arctic soils.

Further, we understand as a criticism from Reviewer 1 that is has been suggested before that including soil fauna in models is important. We would like to point out that we fully agree with that, and do not pretend that we are the first to argue for the importance of soil fauna for soil processes. Instead, what we are postulating in this opinion paper is that not only we should include fauna in models, but we should also include the fact that not all fauna groups are everywhere and, more specifically, that, contrary to temperate environments, important faunal groups are missing from arctic soils. We cannot include these faunal groups in models until we know what effect their introduction will have, which – perhaps with the exception of earthworms for which more data exists – requires further empirical evidence. As we point out, by using recently published papers, this fauna effect could be a lot larger than that of e.g. warming alone. Considering the importance of understanding the future arctic carbon balance for the climate system, we therefore believe that what we present in this manuscript is not only novel but also timely.

- I like the idea of the two different scenarios. However, to me, the scenario 'business as usual' would benefit from a reality check. I think that the scenario is somehow too basic and highly unlikely. Permafrost thawing would not keep things business as usual even if diversity would not be affected. Among other things, I would expect i) colonization of the newly-thawed layers (by fauna but also by plant roots), and ii) an increase in the density of soil organisms (altering rates and synergies).

RESPONSE: We agree and have made this clearer in the manuscript text, notably by renaming the "business as usual" scenario to "state-of-the-art". We have also, in the figure, more clearly adapted rooting depth.

We did not mean that we think the "state of the art" scenario is one of two likely scenarios, on the contrary we present it as a necessary simplification, which should now be improved upon. We also agree that changes in densities or process rates can occur regardless of diversity effects and that those are also accounted for in warming experiment or thaw gradients.

However, large changes are most likely with changes in community composition, as also these are quite -in our opinion- likely to happen, as illustrated in the second scenario. And, as mentioned in the manuscript (and also in the response to Reviewer 1) most studies, such as incubation studies, on which estimating the permafrost-carbon- feedback largely relies, do not account for these changes. These studies may be able to capture vertical changes, such as migration of fauna into newly-thawed soil, but cannot capture northward migration of soil fauna unless this is specifically manipulated. Thus, they implicitly assume a 'state-of-the-art scenario'. This does not mean that we think most earlier studies are wrong or irrelevant, they are obviously very important, but we do hope that more people will be open for the second scenario after reading our paper and will indeed set up experiments taking changes in soil fauna and microbial communities into account.

We have also made this clearer in the manuscript text, it now reads:

"The **first scenario**, which seems to be the theory most studies apply, depicts the 'state-of-the-art' (Scenario 1, Fig. 2b). Here, large-scale and dramatic changes in the belowground environment do not lead to a change in the presence or depth-distribution of soil organisms. This conceptual view assumes that the fate of arctic soil C can simply be predicted by combining *in vitro* incubation

studies, field observations, and modelling of soil and plant responses. Such simplifying assumptions have been paramount in providing estimates of the permafrost carbon feedback (e.g., Koven et al, 2015) and some plant-soil interactions (Keuper et al, 2020). However, while warmer soils alone can increase activity and turnover rates of soil organisms, for example through an increase in density (Dollery et al., 2006), we assume that significant changes in the functional potential of the soil organisms only occur with changes in community composition (Crowther et al., 2019)." in lines 204-212.

- The alleviation of functional limitation by fauna in the MS focuses mostly on the effects by meso- and macro-fauna (it is also easy to imagine) but the capacity of micro-fauna to alter soil biogeochemistry and SOM decomposition could take further attention. I would appreciate some extra details on the microbial functions that are missing and, perhaps, how this could be addressed in new experiments

RESPONSE: We agree that changes in the micro-fauna may be just as important, which is also why we give those as much space as for macro and mesofauna in part 3 "Evidence for alleviation of functional limitation with novel soil organisms". This part is led by lines 166-170 "Not only macro- and mesofauna are absent in permafrost soils, also microbial communities differ from those found in the overlying active layer (Doherty et al., 2020; Johnston et al., 2019; Monteux et al., 2018) and the functional potential of these specialized microbial communities for decomposition processes can also be drastically smaller than that of active layer communities." followed by a whole paragraph about recent studies addressing changes in microbial functions.

We have further added a section focused on insights into how microbial functional limitations may be tested (through microbial community manipulation, essentially expanding on Monteux et al 2020) and which functions may be most immediately relevant to target when considering ecosystem functioning (decomposition through extracellular enzymes, nutrient cycling, production of greenhouse gases) in lines L181-187.

Furthermore, we added the following to section 2: "Although their topsoil counterparts may have varying levels of diversity (e.g., Fierer et al., 2012), permafrost microbial communities typically exhibit low diversity, as they are shaped by strong environmental constraints over long time-scales and extreme dispersal limitation due to their frozen environment (Bottos et al., 2018; Ernakovich et al., 2022).", ll. 88-90.

**Responses to reviewers:**

**Reviewer 1**

This manuscript presents perspective ideas about the future role of soil fauna in northern areas, where permafrost is melting, making large pools of carbon accessible to decomposers. This is obviously an important topic that deserves more attention, efforts so far having been done mostly on the abiotic component of the issue.

Being an important topic, the proposed perspectives have already been presented in the ecological literature and I am afraid that the manuscript only brings little novelty.

For example, a core idea of the MS is that novel traits will bring new ecological functions, but this has been previously proposed (1) and applied to North American boreal regions (2). The fact that soil organisms need to be better integrated in C models has also been highlighted eg (3, 4). Perhaps the most innovative part is the criticism of existing experiments (part 2), but I am not sure that Biogeosciences targets the researchers doing such types of experiments.

**REPLY:**

Thank you for your comments and for agreeing that it is an important topic that deserves more attention. However, we disagree with the notion that this manuscript does not add novelty. Hereto, we want to highlight four things:

The listed references do not focus on the Arctic biome as our perspective paper does. The geographic and environmental settings in the Arctic are unique and extrapolating ecological theory outlined for temperate and boreal ecosystems in North America is not straightforward, notably due to the absence of not only burrowing earthworms but other entire clades or even kingdoms in some arctic soil environments.

Though it is quite well-known in general that soil fauna are important for decomposition rates, the arrival of novel organisms with unique traits in arctic soils has not gotten a lot of attention. To a biogeochemist working in arctic ecosystems, the fact that these traits are not static ecosystem properties (and that they may be present in the future) will be new. This is highlighted by the experimental basis on which many predicted models are based: as we mention in our paper, carbon-cycling feedback predictions from northern soils are largely based on incubations or warming experiments in which the soil faunal community is not manipulated. These are the papers that address carbon source/sink functions of arctic tundra under future climate scenarios and that very clearly are a target audience for Biogeosciences readers, as exemplified by the multiple permafrost incubation or *in situ* studies not accounting for changes in soil fauna published last year in Biogeosciences (e.g., Heffernan et al., 2022; Laurent et al., 2022; Gil et al., 2022; Mauclet et al., 2022; Fischer et al., 2022). Our intention with this perspectives paper is to reach the wider geosciences community who may not be aware of the critical role of particular groups of decomposers in soil processes, or of their absence in arctic soils.

Our focus in this perspectives piece is that we do not yet know how newly-arriving soil fauna will change process rates in northern soils, but that recent, and thus, not previously discussed direct empirical evidence for 'trait decomposition effects' in Arctic soils, suggesting that effects could be large. To be able to improve C models by including soil fauna responses we need quantitative evidence that can only be derived from experimental studies in the relevant environment, which we

advocate for. We believe this topic deserves more attention and hope to spark this attention with this manuscript.

The reviewer #1 is right that there are many studies of earthworm invasions in North America and we also cite such studies. Importantly, that literature is heavily dominated by studies on earthworms and not other soil organisms relevant for the arctic (collembola, bacteria etc). Therefore, it is important that we widen the scope to raise that also other soil fauna can spread to new locations and soil depths with far-reaching ecosystem consequences.

The 4th part proposes two simplistic scenarios, overlooking important mechanisms such as vegetation dynamics and its links with soil fauna, competitive exclusion in the context of tradeoffs between competition and colonization and climate change(5) , interactions network rewiring (6) and so on. It also does not separate short term from long term dynamics C and community dynamics, which can be quite different and interact with fires . The possibilities of non linear behaviors with tipping points is also barely mentioned whereas it is a central question (e.g. the provocative compost bomb hypothesis (7)). I think it would be more reasonable to change part 4 and say that we have no real clues about how it will evolve, but mention a number of mechanisms that might play an important role, based on a more thorough literature search beyond soil organisms (for which the manuscript does a good job) and highlight a few key perspectives that need to be explored, at the interface between environment and ecology (= the scope of this journal).

**REPLY:**

Yes, our scenarios are indeed very much simplified and intentionally so. Our aim is to introduce a framework that it easy to understand and sparking interest in the readers of Biogeosciences. We have, however, re-phrased our first scenario as outlined in the response to the Editor above.

specific comments

L15 perhaps change "missing traits" with "novel traits" ?

**REPLY:** rephrased to 'novel functions' (see our response to reviewer 2 for our revision of the use of 'traits' in this paper).

L 18 "micro-organisms", not "microbes"

**REPLY:** Changed to "(i.e., from bacteria to earthworms)".

L 29 soil property "state", not "property"

**REPLY:** Changed to 'soil feature'.

L33 and and the trait matching between decomposers and ressources  (eg (8))

**REPLY:** Added ", and the matching of traits between decomposers and available resources (Lustenhouwer et al., 2020), " to the sentence.

L40-44 this has been discussed in (9)

REPLY: Added "[Nevertheless], the presence of species functions can also be shaped by glacial history (Mathieu and Davies, 2014), and"

L55 "dispersal" not "dispersion"

REPLY: Changed accordingly.

L 85 see (2)

REPLY: We can add this preprint to our references for statements that earthworms and probably also other soil fauna can be invasive species with potentially large impacts on ecosystem processes.

L 87 see (10) which show temporal dynamics of worms

REPLY: While ref (10) cannot assess migration patterns of earthworms with their approach, it does indeed show that earthworms have survived in Arctic climates, and we can add the paper accordingly.

L176 "projections" not "predictions"

REPLY: Changed accordingly.

L185 : competive exclusion may lead to low biodiversity dominated by few species, your point is not obvious.

REPLY: We are not quite sure what the Reviewer means here. Is it that with competitive exclusion there would be no change in effect traits present in the soil community? This seems unlikely as even with competitive exclusion one could reasonably assume that there would be a shift from primarily stress-tolerant and survival focused species to those with a high functional performance (see for example Crowther et al. 2019).

L195-197 : the point here is the ecological niche (thermal niche), not the distribution (limited by dispersal and ecological niche)

REPLY: Added ", that is their current distribution does not reflect a simple thermal niche" at the end of the next sentence (to add the niche concept here).

L229 the feedback is barely mentioned in the MS

REPLY: We discuss decomposition rates and GHG emissions, i.e. the feedback from northern soils to the global climate, throughout the ms, for example in ll. 32-35, 36-38, 49-54, 60-62, 67, 104-105, 112-114, 118, 125-126, 132-135, 138-139, 145-150, 153-158, 158-162, 163-165, 175-177, 194-195 (referring to the line numbers of the original submission).

L251 really close from the figure in (11) perhaps mention it?

REPLY: Yes, good idea. We will add that this figure is inspired by Crowther et al., but created with updated data sources (see main text) and including currently and future frozen carbon pools – a distinction that we find very important in discussing the potential fate of soil C in the Arctic.

Figure 2 : Sorry, I don't really understand what is the message there.

**REPLY:** This is the figure corresponding to the text part 4, see above. We will rework figure 2 for more clarity in an eventual revision process, as suggested by Reviewer 2.

References cited

1.  D. A. Wardle, R. D. Bardgett, R. M. Callaway, W. H. Van der Putten, Terrestrial Ecosystem Responses to Species Gains and Losses. Science 332, 1273–1277 (2011).

2.  J. Mathieu, J. W. Reynolds, C. Fragoso, E. Hadly, Global worming: massive invasion of North America by earthworms revealed. bioRxiv, 2022.06.27.497722 (2022).

3.  J. Filser, et al., Soil fauna: key to new carbon models. SOIL 2, 565–582 (2016).

4.  M. A. Bradford, et al., Managing uncertainty in soil carbon feedbacks to climate change. Nature Clim Change 6, 751–758 (2016).

5.  S. E. Gilman, M. C. Urban, J. Tewksbury, G. W. Gilchrist, R. D. Holt, A framework for community interactions under climate change. Trends in Ecology & Evolution 25, 325–331 (2010).

6.  G. Woodward, et al., Ecological Networks in a Changing Climate. Advances in Ecological Research 42, 72–138 (2010).

7.  S. Wieczorek, P. Ashwin, C. M. Luke, P. M. Cox, Excitability in ramped systems: the compost-bomb instability. Proceedings of the Royal Society A: Mathematical, Physical and Engineering Sciences 467, 1243–1269 (2011).

8.  N. Lustenhouwer, et al., A trait-based understanding of wood decomposition by fungi. Proceedings of the National Academy of Sciences 117, 11551–11558 (2020).

9.  J. Mathieu, J. T. Davies, Glaciation as an historical filter of below-ground biodiversity, Journal of Biogeography. Journal of Biogeography 41, 1204–1214 (2014).

10.  O. Moine, et al., The impact of Last Glacial climate variability in west-European loess revealed by radiocarbon dating of fossil earthworm granules. Proceedings of the National Academy of Sciences 114, 6209–6214 (2017).

11.  T. W. Crowther, et al., The global soil community and its influence on biogeochemistry. Science 365 (2019).

References cited in our **REPLY:**

Fischer, W., Thomas, C. K., Zimov, N., and Göckede, M.: Grazing enhances carbon cycling but reduces methane emission during peak growing season in the Siberian Pleistocene Park tundra site, Biogeosciences, 19, 1611–1633, https://doi.org/10.5194/bg-19-1611-2022, 2022.

Gil, J., Marushchak, M. E., Rütting, T., Baggs, E. M., Pérez, T., Novakovskiy, A., Trubnikova, T., Kaverin, D., Martikainen, P. J., and Biasi, C.: Sources of nitrous oxide and the fate of mineral nitrogen in subarctic permafrost peat soils, Biogeosciences, 19, 2683–2698, https://doi.org/10.5194/bg-19-2683-2022, 2022.

Heffernan, L., Cavaco, M. A., Bhatia, M. P., Estop-Aragonés, C., Knorr, K.-H., and Olefeldt, D.: High peatland methane emissions following permafrost thaw: enhanced acetoclastic methanogenesis during early successional stages, Biogeosciences, 19, 3051–3071, https://doi.org/10.5194/bg-19-3051-2022, 2022.

Laurent, M., Fuchs, M., Herbst, T., Runge, A., Liebner, S., and Treat, C.: Relationships between greenhouse gas production and landscape position during short-term permafrost thaw under anaerobic conditions in the Lena Delta, Biogeosciences, Preprint, https://doi.org/10.5194/bg-2022-122, 2022.

Mauclet, E., Agnan, Y., Hirst, C., Monhonval, A., Pereira, B., Vandeuren, A., Villani, M., Ledman, J., Taylor, M., Jasinski, B. L., Schuur, E. A. G., and Opfergelt, S.: Changing sub-Arctic tundra vegetation upon permafrost degradation: impact on foliar mineral element cycling, Biogeosciences, 19, 2333–2351, https://doi.org/10.5194/bg-19-2333-2022, 2022.

**bg-2022-215**

**Author(s)**: Blume-Werry et al.,2022

**Title:** Ideas and perspectives: Alleviation of functional limitation by soil organisms is key to climate feedbacks from northern soils

**General comments**

This manuscript is about potential functional alleviation by soil functional groups in arctic soils in the context of permafrost melting. Here, the authors present ideas and perspectives on how soil fauna may impact the decomposition of carbon sources made available by melting permafrost. This topic deserves more attention, particularly since it could be applied on a larger scale by emphasizing the importance of the ecosystem's multifunctionality. Furthermore, the resilience of above- and belowground ecosystems is essential when discussing potential feedback loops regarding greenhouse gas emissions. Overall, I found the manuscript well written, clearly highlighting the major gaps in this field, and also proposing an interesting roadmap for future research. Below, I suggest venues for minor improvements:

1) In this paper, the authors present two different scenarios according to which artic soil communities are impacted and acted upon differently by abiotic factors such as temperature and community changes (scenario 1), and that the functional diversity of these communities is different and that functional trait limitation of these species is likely to be responsible for decomposition dynamics (scenario 2). Both scenarios are straightforward but could better present the effects of multifunctionality on the dynamics of organic matter decomposition. For instance, by underlining the importance of permafrost melting (i.e., artic environments) while linking the different functions played by the different groups of soil organisms and their impact on the functional ecology of these environments. For instance, additions could be made concerning the metabolic profile of soil microbial communities for specific carbon sources consumptions. Doing this could lead to complementary ideas and new perspectives to conclude with investigate, especially regarding experimental designs used and in this field.

REPLY:

If offered a chance to revise our manuscript, we will include a discussion about the complexity of organic matter decomposition (see also our reply to Reviewer #1). When it comes to dynamics of microbial metabolism, we believe that discussing the complex interplay between microbial dispersal, differing functional potential, coalescence or other community assembly processes, substrate availability and redox status would become overly specialized for an opinion piece. However, we have now added a short piece about these thematics as avenues for future research (ll. 182-187, revised manuscript), see our response to the Editor above.

**Specific comments:**

Figure 2 could be enlarged (panels a & b), and the headers (panel b) could be centered for better readability.

REPLY: The size of the figure will of course depend on the typesetting of the journal, but we are happy if the figures are larger than they are now. We also changed the headings as suggested, see also our answer below.

**L 9**    Abstract and keywords to put in alphabetical order?

REPLY: Done.

**L16**    Maybe list/name of or more traits that could alleviate functional limitation?

REPLY: Yes, we changed this to: "…may introduce 'novel functions', resulting in increased rates of *e.g.*, nitrification, methanogenesis, litter fragmentation, or bioturbation,…"

**L36**    Clearly differentiate organisms' "traits" and "properties"

REPLY: Yes, this is not necessarily straightforward. Often the 'traits' themselves are not missing, but the trait 'value' that would have an effect on functions is missing. For example, some litter-dwelling earthworms may be present but the deeper burrowing species missing. We realised that it would be better to not use the term 'trait' to avoid confusion and are now using soil organism *functions* instead.

**L37**    bioturbation and litter fragmentation are not traits or either properties but more processes.

REPLY: yes, we have now changed this (in accordance to your comment above also). It now reads:

"We here refer to this idea as 'functional limitation' and use 'soil organism functions' as the direct effect of an organisms' activity (via its combined functional or 'effect' traits) on soils, litter, or substrate (e.g. the ability or extent to which they perform nitrification, methanogenesis, litter fragmentation, microbivory, or bioturbation)."

**L59**    What do you mean by **"**well by vertical (downward) dispersal of missing soil organisms". This sentence might need additional information and clarification.

REPLY: Yes, there was a word missing. We changed this to "as well as by vertical (downward) dispersal of novel soil organisms"

**L64**    "settle" instead of "arrive"?

REPLY: Changed as suggested.

**L84**    How might the permafrost open up new niches? Perhaps this idea could be expanded a bit more for clarification.

REPLY: Expanded with: "as the physical barrier of frozen soils is removed and thus far fauna-free soils can be colonized"

**L182-183** What about the effect of topography/ elevation?

REPLY: We are unfortunately not sure what is meant here.

**208-210** "… That is, there might not….", not sure if this sentence is clear, rephrase it?

REPLY: Changed to: "This means that there might not only be more species with new properties in the topsoil but also…"

**L252-265** Figure title is too long, reformulate for clarity, maybe something like: "*Latitudinal distribution of soil carbon above- and belowground biomass and functionally limited arctic soil food web according to two current scenarios*. …".

REPLY: We have shortened the figure legend, and added to b), as suggested below

**Figure 1**: mention that this figure is inspired by Crowther TW et al. 2019?

REPLY: You probably mean Fig. 2a. We will add that this figure is inspired by Crowther et al., but created with updated data sources and including currently and future frozen carbon pools, see also our response to Reviewer #1.

**Figure 2a**: prefer a description of the figure in the manuscript to a description in the figure's legend itself, this is unclear and needs to be clarified. It is essential to mention the data sources used to graph these results, but this should be part of the manuscript, for instance, when discussing used experimental design in arctic environments (see part 4).

REPLY: Yes, we have moved this to the beginning of part 4.

**Figure 2b**: here, the legend describes the figure well, but some information is missing:  an introductory sentence briefly describing what panel b shows,

and some indicators: maybe write "Current food web" and not only "current," then center the text of each heading for more readability. Numbered each box could also be done to optimize the understanding of the legend (I= current food web, II= "business as usual" future scenario 1, III= "functional alleviation," future scenario 2  The greyed part illustrating the permafrost could be colored blue, as in figure 1.

REPLY: We added the introductory sentence "Conceptual overview of the current food web and two different future scenarios." to the figure legend, and changed the figure as suggested.

**Second round of revisions**

**Reviewer #1**

Accept as is.

**Reviewer # 2**

Suggestions for revision or reasons for rejection

(visible to the public if the article is accepted and published)

Soil organisms are apparently key to understanding and predicting future climate feedbacks from permafrost soils in the Arctic, where large amounts of carbon store. This review proposed that the warming-induced arrival of soil organisms into arctic soils may introduce 'novel functions', increases rates of e.g., nitrification, methanogenesis, litter fragmentation, and thereby alleviates functional limitations of the current community. Therefore, the dispersal of until-then absent micro-, meso- and macro-organisms (i.e., from bacteria to earthworms) into new regions and newly-thawed soil layers can drastically affect carbon cycling in arctic soils.

I understand this is a revised version, although I did not review its early versions. Generally, this review is very interesting, which could bring novel ideas into the scientific community. The manuscript is well structured and written, and I was very happy to read it. I only have a concern about the microbial arrival into the arctic soils. Section 2, this section is about the disperse of soil organisms in arctic soils, which should include fauna and microbes, whereas only faunas were discussed. I would encourage authors to include the microbial disperse into the arctic soils, e.g. how and what microbes move to the arctic soils under the warming scenario. This microbial disperse ways could be more complicated than faunas, e.g. vertical, horizon or even microbial banks in the frozen soils.

Thank you for the positive assessment and the suggestion for improvement.

In line with the suggestion of Reviewer #2, we have added the following text to section 2 (ll. 115-124):

"Microbial dispersion into Arctic soils functions in a different way than faunal dispersal. Here, lateral, northward dispersal is likely less limiting than for soil fauna because airborne dispersal is widespread in many bacteria and fungi (Harding et al., 2011; Thompson et al., 2017), but the vertical dispersal of micro-organisms into newly-thawed layers is the subject of ongoing investigations. While microbial communities in newly-thawed permafrost can converge with those observed in the active layer (e.g. Monteux et al., 2018; Doherty et al., 2020), it remains unclear whether this stems from microbial migration downward or from modifications of the existing permafrost microbial community. What further complicates predictions of future microbial communities in the Arctic is the existence of ancient bacteria and viruses in permafrost which, after being dormant for millennia in frozen soil layers, can become active again upon thaw (Miner et al., 2021). How the active and dormant microorganisms currently present in permafrost will interact with newly-arriving microorganisms to determine the assembly of post-thaw permafrost microbial communities is still unclear (Ernakovich et al., 2022)."